

# Global atmospheric budget of simple monocyclic aromatic compounds

David Cabrera-Perez[1], Domenico Taraborrelli[1,2], Rolf Sander[1], and Andrea Pozzer[1]

[1]Atmospheric Chemistry Department, Max-Planck Institute of Chemistry, Hahn-Meitner-Weg 1, D-55128 Mainz, Germany
[2]now at: Institute of Energy and Climate Research (IEK-8), Forschungszentrum Jülich GmbH, Jülich, Germany

*Correspondence to:* D. Cabrera-Perez (d.cabrera@mpic.de)

**Abstract.** The global atmospheric budget and distribution of monocyclic aromatic compounds is estimated, using an atmospheric chemistry general circulation model. Simulation results are evaluated with an ensemble of surface and aircraft observations with the goal of understanding emission, production and removal of these compounds.

Anthropogenic emissions represent the largest source of aromatics in the model ($\simeq 25$ Tg/yr) and biomass burning the second largest ($\simeq 6$ Tg/yr). The chemical production of aromatics accounts for $\simeq 6$ Tg/yr. The atmospheric burden of aromatics sums up to 0.3 Tg. The main removal process of aromatics is photochemical decomposition ($\simeq 32$ Tg/yr), while wet and dry deposition are responsible for a removal of $\simeq 4$ Tg/yr.

Simulated mixing ratios at the surface and elsewhere in the troposphere show good spatial and temporal agreement with the observations for benzene, although the model generally underestimates mixing ratios. Toluene is generally well reproduced by the model at the surface, but mixing ratios in the free troposphere are underestimated. Finally, larger discrepancies are found for xylenes: surface mixing ratios are not only overestimated but also a low temporal correlation is found with respect to in situ observations.

## 1 Introduction

Volatile Organic Compounds (VOCs) play a significant role in the chemistry of the troposphere and in ozone formation (Atkinson, 2000; Seinfeld and Pandis, 2012). Within the VOCs class, aromatic compounds form a subgroup of special interest: in the troposphere of urban and semi-urban areas, aromatic hydrocarbons comprise a major fraction (up to 60%) of the VOCs (Lee et al., 2002; Ran et al., 2009). Consequently, they are highly relevant for ozone formation in these areas (Kansal, 2009; Barletta et al., 2005; Koppmann, 2008) as they can be responsible for up to 50% of the total ozone formation potential (Tan et al., 2012). Even in rural areas, high levels of aromatics have been reported, summing up to 35% of the total VOCs (Guo et al., 2006; You et al., 2008).

Typical benzene and toluene mixing ratios fall within the 0.1-10 pmol/mol range. Estimated lifetimes are 2 days for toluene and 2 weeks for benzene (Koppmann, 2008). These lifetimes are long enough to allow the compounds to reach downwind areas far from sources, as for instance the Sahara desert (Yassaa et al., 2011).



**Table 1.** Aircraft and surface measurements of aromatics that are used for comparison with model simulations.

| Site name | Time | Location | Reference |
|---|---|---|---|
| Surface measurements | | | |
| Karachi | Dec 1998 - Jan 1999 | Pakistan | (Barletta et al., 2002) |
| 43 cities | Jan-Feb 2001 | China | (Barletta et al., 2005) |
| 28 cities | 1999-2005 | US | (Baker et al., 2008) |
| 12 sites | 1996 | UK | (Derwent et al., 2000) |
| Rome | 1992-1993 | Italy | (Brocco et al., 1997) |
| Antwerp | Sept 2003 - Oct 2005 | Belgium | (Buczynska et al., 2009) |
| Paris | 2010 | France | (Dolgorouky et al., 2012) |
| Kolkata | March - June 2006 | India | (Dutta et al., 2009) |
| Pear River Delta | Aug 2001 - Dec 2002 | China | (Guo et al., 2006) |
| Boreal forest | Apr 2000 - Apr 2002 | Finland | (Hakola et al., 2003) |
| Mount Tai | June 2006 | China | (Inomata et al., 2010) |
| Welgegund | Feb 2011 - Feb 2012 | South Africa | (Jaars et al., 2014) |
| Oil refinery | 1997 | Greece | (Kalabokas et al., 2002) |
| Kathmandu | Jan-Feb 2003 | Nepal | (Yu et al., 2008) |
| Algiers | Nov 1999 | Algeria | (Kerbachi et al., 2006) |
| Rio de Janeiro | Jan 2001 | Brazil | (Martins et al., 2007) |
| Windsor | 2004-2006 | Canada | (Miller et al., 2012) |
| Delhi | Oct 2001-Sept 2002 | India | (Hoque et al., 2008) |
| Ankara | Jan - June 2008 | Turkey | (Yurdakul et al., 2013) |
| Bangkok | July 2008 | Thailand | (Suthawaree et al., 2012) |
| 6 sites | Nov 1999 | Sahara desert | (Yassaa et al., 2011) |
| EMEP | 2005 | Europe | (Tørseth et al., 2012) |
| EEA | 2005 | Europe | http://www.eea.europa.eu/data-and-maps/data/airbase-the-european-air-quality-database-8 |
| Aircraft measurements | | | |
| Caribic aircraft (∼ 11 km) | 2005-2012 | global/multiple locations | (Baker et al., 2008) |

Aromatic VOCs are emitted by a range of sources. They form a relevant fraction of fossil fuels, and they are released into the atmosphere by combustion (i.e., gasoline and diesel engines), gasoline evaporation, solvent usage, and spillage (Sack et al.,





1992; Kim and Kim, 2002; Na et al., 2004; Baek and Jenkins, 2004). In urban air masses benzene, toluene, ethylbenzene, xylenes, styrene and trimethylbenzenes are highly present (Koppmann, 2008). After anthropogenic emissions, biofuel and biomass burning are the second largest sources of aromatics. They are important sources of benzene, toluene and phenol in tropical and boreal areas (Fu et al., 2008; Henze et al., 2008; Andreae and Merlet, 2001). Finally, only toluene is considered

to be biogenically emitted (Sindelarova et al., 2014), although a recent study pointed out that biogenic emissions of simple aromatics could be equal in importance to anthropogenic emissions (Misztal et al., 2015).

The primary atmospheric oxidation pathway of benzene and alkyl-substituted benzenes is via the reaction with OH, followed by the reaction with $NO_3$ (Atkinson, 2000, and references therein). The oxidation products of aromatic compounds contribute to ozone formation and production of secondary organic aerosol (SOA) (Odum et al., 1997; Butler et al., 2011).

Besides, there is a variety of chemical processes in the atmosphere that involve aromatic oxidation products, which can influence OH recycling in the atmosphere. For instance, *ortho*-nitrophenols are species of interest due to their HONO-production upon photolysis (Bejan et al., 2006; Chen et al., 2011). Moreover, nitrophenols are emitted directly into the atmosphere by traffic and biomass burning (Tremp et al., 1993; Mohr et al., 2013).

Many aromatic compounds can be dangerous for humans, animal life and plants (Ciarrocca et al., 2012; Snyder et al., 1993).

For instance, benzene is known to be carcinogenic (Snyder et al., 1993); toluene and xylenes can have severe effects on the neural system (WMO, 2000; Sarigiannis and Gotti, 2008); and nitrophenols have acute toxicity for humans and plants (Natangelo et al., 1999; Michałowicz and Duda, 2007). Due to the high noxiousness and atmospheric impacts, aromatics have been subject of monitoring and measurement campaigns (see Table 1), aiming at establishing control strategies for environmental and human health protection.

For these reasons, it is important to have a correct model description of aromatic compounds in the atmosphere, and to have a detailed knowledge of their budget, as this will improve our understanding of their photochemical production yields (Lewis et al., 2000).

However, so far only a few regional or global scale studies focused on aromatics (Henze et al., 2008; Hu et al., 2015; Lewis et al., 2013), while most of the global studies on VOCs only focused on aliphatic hydrocarbons (Millet et al., 2010; Pozzer

et al., 2010; Fu et al., 2008; Paulot et al., 2011; Fischer et al., 2014, e.g.). To our knowledge, this is the first comprehensive atmospheric budget study on major monocyclic aromatics in the gas-phase.

This work focuses on the gas phase chemistry of simple aromatics, hence neglecting any SOA production. Other global studies as Henze et al. (2008) did not omit SOA production although they were focused on the aerosol phase.

In this work we focus only on the most abundant simple aromatics: benzene, toluene, xylenes, phenol, styrene, ethylbenzene,

trimethylbenzenes and benzaldehydes. The purpose of this study is to: (1) simulate the aromatic compounds in the atmosphere with an atmospheric chemistry general circulation model; (2) evaluate the model results by comparing different simulated emission scenarios with atmospheric observations from a number of surface and aircraft campaigns and monitoring stations; (3) provide an estimate of the global atmospheric budget of the most abundant aromatic species in the atmosphere.

In section 2, we present the model setup, including a detailed description of the chemical oxidation mechanism of the

aromatic compounds, emissions and sinks. A detailed description of the observations used for the model evaluation is given in



**Table 2.** Total annual anthropogenic emissions for the different emission scenarios in Tg/yr.

| Scenario | Benzene | Toluene | Xylenes | reference |
|----------|---------|---------|---------|-----------|
| *RCP* | 6.88 | 7.32 | 6.25 | (Lamarque et al., 2010) |
| *LIT* | 3.51 | 6.00 | 4.56 | (Fu et al., 2008) |

section 3. A comparison of model results with the observations is shown in Sect. 4. Finally, we discuss the atmospheric budget of aromatic compounds in terms of sources, sinks and spatial distribution (Sect. 5), followed by the conclusions and outlook.

## 2 Model description and setup

We use the ECHAM/MESSy Atmospheric Chemistry (EMAC) model[1]. EMAC is a numerical chemistry and climate simulation system which includes the 5th generation European Centre Hamburg general circulation model (ECHAM5, Roeckner et al. (2006))as the core atmospheric model. Several submodels describing atmospheric processes are connected via the Modular Earth Submodel System (MESSy2.50) (Jöckel et al., 2010).

For this study, we use the T63L31ECMWF resolution, which corresponds to a horizontal resolution of T63 with spherical truncation (i.e., a Gaussian grid of approx. $1.9° \times 1.9°$ in latitude and longitude). In this setup, the model has 31 vertical hybrid pressure layers up to 10 hPa. The simulation was nudged towards ECMWF analysis data for a realistic representation of tropospheric meteorology (Jeuken et al., 1996). In order to have the same atmospheric dynamics in all sensitivity simulations, the feedback between chemistry and dynamics is switched off (Chemical Transport Model mode (Deckert et al., 2011)).

We performed a 24-month simulation from January 2004 to December 2005. The first 12 months are used as spin-up, and only the results for 2005 are used for the analysis.

Two different scenarios have been simulated, differing only with respect to anthropogenic emissions. A detailed description of the scenarios can be found in the following section. A summary of the scenarios and the emissions of the different species can be found in Table 2.

In addition, box model simulations have been performed in order to better understand the chemical mechanism used in this work.

### 2.1 Emissions of aromatics

**Anthropogenic**

Emissions of aromatics are primarily anthropogenic, coming from numerous sources, including fuel evaporation and combustion, spillage, solvent use, refining of gasoline, landfill wastes and coal-fired stations (Kansal, 2009).

---

[1] http://www.messy-interface.org





**Table 3.** Biomass burning emission factors for the BIOBURN submodel. Emission factors are given in units of g/kg.

| Specie | Savanna | Tropical forest | Boreal forest | agriculture | peat | reference | emissions(Tg/yr) |
|---|---|---|---|---|---|---|---|
| Benzene | 0.20 | 0.39 | 1.11 | 0.15 | 1.21 | (Akagi et al., 2011) | 1.60 |
| Toluene | 0.08 | 0.26 | 0.48 | 0.19 | 2.46 | (Akagi et al., 2011) | 1.00 |
| Xylenes | 0.05 | 0.11 | 0.18 | 0.01 | 0.00 | (Andreae and Merlet, 2001; Akagi et al., 2011) | 0.32 |
| Phenol | 0.52 | 0.45 | 0.48 | 0.52 | 4.36 | (Akagi et al., 2011) | 2.43 |
| Styrene | 0.02 | 0.03 | 0.13 | 0.03 | 0.00 | (Andreae and Merlet, 2001) | 0.16 |
| Ethylbenzene | 0.13 | 0.02 | 0.05 | 0.03 | 0.00 | (Andreae and Merlet, 2001) | 0.08 |
| Trimethylbenzenes | 0.00 | 0.00 | 0.12 | 0.00 | 0.00 | (Yokelson et al., 2013) | 0.06 |
| Benzaldehyde | 0.03 | 0.03 | 0.04 | 0.01 | 0.00 | (Andreae and Merlet, 2001; Yokelson et al., 2013) | 0.12 |

In our study, emissions due to human activities are taken from the Representative Concentration Pathways (RCP) inventory (Van Vuuren et al., 2011). The RCP dataset was used in the IPCC's Fifth Assessment Report and consists of a set of four emission scenarios, developed by four different modeling groups (van Vuuren et al., 2008). Each scenario has a specific radiative forcing for the year 2100 (2.6, 4.5, 6.0, and 8.5 $Wm^{-2}$) (Van Vuuren et al., 2011). We selected the RCP8.5 pathway (Riahi et al., 2007). Granier et al. (2011) indicate that this assumption is reasonable for the time span 2000-2010. The dataset has a yearly resolution and no seasonal variation. We adopted a vertical distribution of emissions based on the work of Pozzer et al. (2009).

Two simulations with identical meteorology and different anthropogenic emissions have been performed. One scenario has the default emissions developed by the IPCC (denoted as *"RCP"*). The second scenario, called *"LIT"*, has scaled *RCP* emissions, which are adapted to reproduce the total annual anthropogenic emissions reported by Fu et al. (2008). A summary of the scenarios and their total emissions for the different species can be found in Table 2. The scenarios are only different for benzene, toluene and xylenes, since no literature studies have been found for other species. Both scenarios are compared with surface and tropospheric observations, as described in Sect. 4.

In the *RCP* simulation, 25 Tg of aromatics are released into the atmosphere, which represents 18% of the total anthropogenic VOC emissions. In the *LIT* scenario, 18 Tg are emitted, and the aromatics represent 13% of the total anthropogenic VOC emissions.

**Biomass burning**

Biomass burning presents a large source of VOCs for the atmosphere (Lamarque et al., 2010). This contribution is represented by the BIOBURN submodel. BIOBURN calculates the emission fluxes, based on the Global Fire Assimilation System (GFAS) datasets (Kaiser et al., 2012). GFAS uses satellite retrievals of fire radiative power and transforms these into dry matter combustion rates. GFAS has a daily time resolution, and therefore seasonal variations can be observed. The dry matter combustion rates are used in the model in combination with biomass burning emission factors to estimate the biomass burning emissions of specific compounds.





For each aromatic species, we applied emission factors retrieved from literature (Yokelson et al., 2013; Andreae and Merlet, 2001; Akagi et al., 2011). The emission factors used in this work are listed in Table 3. For other VOCs, we selected evaluated factors as in Pozzer et al. (2010).

Akagi et al. (2011) showed that at least 400 Tg/yr of VOCs are emitted into the atmosphere from biomass burning. Approximately 6 Tg/yr are aromatics, which consequently represent less than 2% of the total biomass burning VOC emissions. It is worth mentioning the study of Johnson et al. (2013), who estimated an emission of 13.31 Tg/yr for phenol. These emissions are dominated by open cooking, although it remains unclear how calculations were done. In the present study open cooking emissions are included within anthropogenic sources but the *RCP* database does not present such amount of emissions.

**Biogenic**

Biogenic emissions have been reported for more than 25 aromatic species (Misztal et al., 2015), although at low amounts. Moreover, most compounds have complex structures (polycyclic aromatics) or more than 8 carbon atoms. These compounds are out of the scope of this paper, since only emissions of simple monoaromatic compounds, such as toluene, have been considered here. Toluene fluxes from plants have been measured. The production mechanism is not clear (Heiden et al., 1999) but a considerable source of toluene from vegetation of more than 1 Tg/yr has been reported (Sindelarova et al., 2014). In this study, biogenic emissions are calculated online by the MEGAN model (v2.04, Guenther et al. (2006)). For toluene, the model yields an emission rate of $\simeq 0.35$ Tg/yr.

## 2.2 Chemistry

Reaction with OH is the major removal process of aromatic compounds in the troposphere, followed by a small contribution via reaction with $NO_3$ radicals (Atkinson, 2000). For benzene and the alkyl-substituted benzenes there are two possible pathways concerning the OH reaction, the first and most prominent is the OH radical addition, which amounts to about 90% of the reactions, and the other 10% correspond to H-atom abstraction (Atkinson, 2000). Only for styrene, which contains a non-aromatic double bond, the reaction with $O_3$ is relevant. Phenol undergoes mostly OH-addition, while benzaldehyde reacts almost exclusively via H-abstraction (Clifford et al., 2005).

In our model, chemical kinetics calculations are done with the MECCA submodel (Sander et al., 2011) which uses the Kinetic PreProcessor (KPP, Sandu and Sander (2006)). For this study, a new reaction mechanism for aromatics has been developed and added to MECCA (Taraborrelli et al, manuscript in preparation). It describes the chemistry of benzene, toluene, xylenes, phenol, styrene, ethylbenzene, trimethylbenzenes, benzaldehydes and higher aromatics (lumped alkyl-substituted benzenes with 10 or more carbon atoms). The new scheme is based on the Master Chemical Mechanism (MCMv3.2), the most detailed oxidation mechanism available for aromatics with 3788 reactions and 1271 species (Bloss et al., 2005b). However, since the MCM is too computationally expensive for global models, it had to be reduced to 666 reactions and 229 species. A complete list of chemical equations and species involved can be found in the electronic supplement (doi:10.5194/acp-0-1-2016-supplement). The mechanism reduction has been performed according to the following procedure:





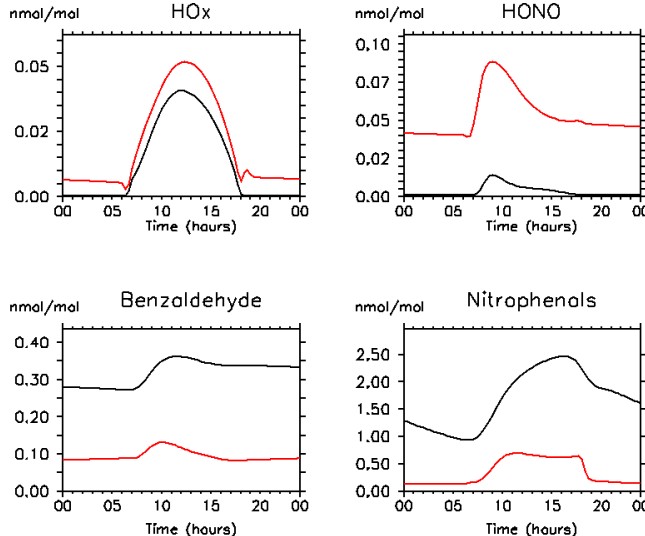

**Figure 1.** Mixing ratios from box model simulation. In red, mixing ratios from the mechanism used in this work. In black, mixing rations from the same mechanism without the updated photolysis rates for benzaldehyde and the new photolysis channels for nitrophenols.

1. The oxidation mechanisms for benzene and toluene were taken from MCM, because of their relatively high abundance in the atmosphere in comparison with the other aromatics. Therefore, these two species are described with the highest available accuracy.

2. For the other aromatics, the first oxidation step is taken from the MCM, and the second oxidation step is linked to that of toluene, because of the similar chemical structure.

3. Intermediates having a lifetime always below one second are replaced by their products with the corresponding reactions being removed.

4. Xylenes and trimethylbenzenes have been lumped, assuming equal proportions of single isomers.

Moreover, the photolysis rate of benzaldehyde is updated following IUPAC recommendations (http://iupac.pole-ether.fr) and *ortho*-nitrophenols have a new photolysis channel leading to HONO formation.

Although shorter or more simplified mechanisms of aromatic decomposition already exist (CRI, MOZART-4) (Jenkin et al., 2008; Emmons et al., 2010), the chemical degradation scheme introduced here allows for the introduction of new features, for instance the photolysis of nitrophenols and updated rate constants. In contrast, CRI and MOZART-4 are less explicit and more difficult to extend because of the high number of lumped species that they contain.

In order to better understand which are the atmospheric implications for the global simulations due to the developed mechanism, we run two simulations with the CAABA/MECCA box model (Sander et al., 2011). The simulations are representative





for the summer period in urban equatorial areas and include emissions for benzene, toluene, phenol and NO (the details of the setup can be found in the doi:10.5194/acp-0-1-2016-supplement). Figure 1 shows the differences in mixing ratios between two box model simulations: one uses the mechanism employed in the global simulations (i.e. the MCM mechanism reduced and updated as explained in this section). The second simulations uses the same mechanism without any of the photolysis updates mentioned above. Benzaldehyde mixing ratios for the updated version of the mechanism are lower than the non-updated version because the latest photolysis rate from IUPAC is faster than the previous one, leading to an increase in the production rate of nitrophenols. Despite this increase in the production, nitrophenols are almost depleted because the new photolysis channel is included, revealing the strong influence of the photolysis channel as a sink. HONO is enhanced in the updated version due to formation via photolysis of nitrophenols. Consequently, HOx (OH+HO$_2$) production has also increased due to HONO recycling and OH formation.

Atmospheric oxidation of some aromatic compounds can result in a major production (or sink) of other (aromatic) compounds. For instance, when benzene reacts with OH, more than 50% is transformed to phenol. Due to the large amount of benzene that is released into the atmosphere, up to 4 Tg/yr of phenol is produced by this pathway, which represents more than 50% of the global phenol source (see Table 5). Additionally, benzaldehyde is produced from several oxidation pathways, which together constitute more than 50% of the total benzaldehyde source.

## 2.3 Sinks

### Scavenging and dry deposition

Because of the hydrophobic nature of aromatic hydrocarbons, scavenging is a minor sink in the atmosphere. However, most oxidation products of aromatics have a strong hydrophilic character. Therefore, wet deposition is a removal process of minor importance for aromatics but essential for its oxidation products.

In the model, dry deposition velocities are calculated using an algorithm based on the big leaf approach (Wesely, 1989; Ganzeveld and Lelieveld, 1995). To account for scavenging, the aqueous and gas-phase chemistry is coupled with physical processes related to clouds and precipitation, which represents the wet deposition (Tost et al., 2006).

Scavenging and dry deposition are calculated in the MESSy submodels SCAV (Tost et al., 2006) and DDEP (Kerkweg et al., 2006), respectively. The Henry's law constants used in these calculations are listed in the electronic supplement (doi:10.5194/acp-0-1-2016-supplement).

## 3 Observations

To evaluate the model simulations (Sect. 4), we collected a set of observations from aircraft and surface campaigns, and from monitoring stations. Table 1 summarizes the locations and periods of the different campaigns.

Observations include data from:





- **CARIBIC**: The CARIBIC project (Civil Aircraft for the Regular Investigation of the atmosphere Based on an Instrument Container) is a long term monitoring program, based on atmospheric measurements on board of a passenger aircraft (Lufthansa A340-600) (Brenninkmeijer et al., 2007; Baker et al., 2010). Cruising altitudes are 10-12km and, on average, 50% in the upper troposphere and 50% in the lowermost stratosphere. The data spans from 2005 to 2012. CARIBIC flights take off from Frankfurt (Germany) on routes to India, East Asia, South America and North America.

- **EMEP**: The European Monitoring and Evaluation Programme (EMEP) (Tørseth et al., 2012) is a network of monitoring sites over Europe and includes measurements of a wide number of species. One of its principal aims is to quantify the long-range transmission of air pollutants, and their fluxes across boundaries. The Chemical Coordinating Centre at NILU (Norwegian Institute for Air Research) is responsible for the data harmonization after the data has successfully passed quality controls. EMEP sites are located in such way that local influences are minimal, and consequently the observations are representative of large regional areas. The data were downloaded from http://www.nilu.no/projects/ccc/emepdata.html.

- **EEA**: Data provided by the European Environmental Agency (EEA) are based on the public air quality database AirBase. EEA gathers information from a large network of monitoring stations in urban, semi-urban and background areas. However, only rural background stations are used for the comparison because the simulation horizontal scale is not representative for traffic or industrial influenced stations. Moreover, for the comparison, annual averages for each station have been used. We selected observations from the year 2005. However, the number of stations that is feasible for this study is small. The data were downloaded from http://www.eea.europa.eu/data-and-maps/data/airbase-the-european-air-quality-database-8.

- **Literature**: A compilation of measurements from the literature, covering multiple parts of the globe in multiple campaigns, is summarized in Table 1. The table provides detailed information on the location and the time span of the observational dataset. The data covers the years 1995-2012 for 111 surface sites located in rural, semi-urban and urban areas. Each observation represents a different period, ranging from months to years (e.g., Barletta et al. (2005)), which can be a source of error in the comparison.

## 4 Evaluation with observations

In this section the model results for benzene, toluene and xylenes are evaluated by comparison with observations. Comparisons for other aromatic compounds cannot be made due to lack of a consistent set of global measurements. The full set of figures can be found in the electronic supplement (doi:10.5194/acp-0-1-2016-supplement).

Model results for the year 2005 were chosen for the comparison with observations, assuming that interannual variability is not a significant source of error and that emissions of aromatics were rather constant over the the period 1995-2010 for the *RCP* dataset with a relative increase of 3% .





Table 4 summarizes the statistics of the model-measurement comparison for the 3 species mentioned above and for the monitoring networks described in section 3. To calculate the statistics, model results were sampled within the geographical locations of the observations.

We follow the criteria of Barna and Lamb (2000) and Pozzer et al. (2012) for the analysis of the model performance. A ratio
of RMS (root mean square error) to STD (standard deviation of observations) below 1 is taken as the criterion to establish good modelling quality. In general, the ratio RMS/STD gives a better agreement for the *RCP* simulation than for the *LIT* simulation, but both simulations give ratios close to one, meaning relatively good agreement. As a final note, comparisons with station observations must be taken cautiously, as they could be influenced by local emissions and consequently not fully representative for background air, which would be best suited for comparison with large scale models as the one used in this work.

**Table 4.** Summary of the statistical comparison of observed and simulated mixing ratios. Arithmetic means and standard deviations are shown in pmol/mol. M*LIT*, M*RCP* and MObs represent the mean values for the *LIT* simulation, the *RCP* simulation and the observations, respectively. S*LIT*, S*RCP* and SObs are the standard deviations of the previously mentioned cases.

| Species | Network | Number of locations | Time resolution | M*LIT* | S*LIT* | M*RCP* | S*RCP* | MObs | SObs | MObs /M*LIT* | MObs /M*RCP* | RMS (*LIT*) | RMS (*RCP*) | *RCP*: RMS /SObs | *LIT*: RMS /SObs |
|---|---|---|---|---|---|---|---|---|---|---|---|---|---|---|---|
| Benzene | CARIBIC | 1241 | Instantaneous | 6.6 | 2.1 | 13.4 | 3.9 | 16.0 | 15.8 | 2.45 | 1.20 | 18.0 | 15.5 | 1.0 | 1.1 |
| | EEA | 22 | Annual mean | 76.4 | 11.0 | 144.4 | 23.4 | 194.0 | 118.4 | 2.54 | 1.34 | 167.1 | 129.4 | 1.1 | 1.4 |
| | EMEP | 14 | Monthly | 84.0 | 35.9 | 158.5 | 65.9 | 213.6 | 97.7 | 2.54 | 1.35 | 139.2 | 91.9 | 0.9 | 1.4 |
| | *LIT*ERATURE | 105 | Campaign dependent | 205.6 | 173.8 | 393.2 | 339.5 | 1500.0 | 1968.0 | 7.30 | 3.81 | 2301.0 | 2168.0 | 1.1 | 1.2 |
| Toluene | CARIBIC | 789 | Instantaneous | 0.7 | 0.5 | 0.8 | 0.6 | 3.6 | 7.5 | 5.22 | 4.33 | 8.0 | 7.9 | 1.1 | 1.1 |
| | EEA | 6 | Annual mean | 251.9 | 9.2 | 306.3 | 11.2 | 240.3 | 59.4 | 0.95 | 0.78 | 54.2 | 83.8 | 1.4 | 0.9 |
| | EMEP | 11 | Monthly | 183.9 | 54.0 | 270.2 | 78.7 | 137.3 | 37.1 | 0.75 | 0.51 | 128.6 | 188.9 | 5.1 | 3.5 |
| | LITERATURE | 105 | Campaign dependent | 290.3 | 200.3 | 352.1 | 243.4 | 2454.0 | 3499.0 | 8.45 | 6.97 | 4100.0 | 4066.0 | 1.2 | 1.2 |
| Xylenes | EMEP | 9 | Monthly | 114.8 | 34.5 | 156.8 | 47.0 | 88.0 | 6.2 | 0.77 | 0.56 | 57.7 | 85.5 | 13.7 | 9.3 |
| | LITERATURE | 53 | Campaign dependent | 120.7 | 78.1 | 164.8 | 106.8 | 2026.0 | 6149.0 | 16.79 | 12.29 | 6452.0 | 6445.0 | 1.0 | 1.0 |

## 4.1 Benzene

*EEA:* This set of 22 stations with observations for 2005 shows annually averaged mixing ratios of 194 nmol/mol (see Table 4). In general, the model underestimates observations by a factor of 2.5 in the *LIT* simulation and by 0.45 in the *RCP* simulation. As expected due to the coarse model resolution, the simulated spatial variability of simulations (with standard deviations of 11 and 23 pmol/mol for *LIT* and *RCP*, respectively) is lower than that of the observations (118 pmol/mol). The RMS shows better
agreement for the *RCP* than for the *LIT* simulation. In addition, *RCP* and *LIT* show good spatial agreement for all stations, except for one station in central Europe (figures can be found in the supplement, doi:10.5194/acp-0-1-2016-supplement). In conclusion, this comparison suggests that emissions from the *RCP* scenario give better results in Europe.

*EMEP:* This dataset has a daily resolution for 14 stations located in Europe. In this work only monthly values estimated from the database are used. In Fig. 2, the *RCP* simulation results for benzene are compared with observations from six stations.
It can be noticed that mixing ratios are better captured by the model in summer than in winter, which is a feature that has been




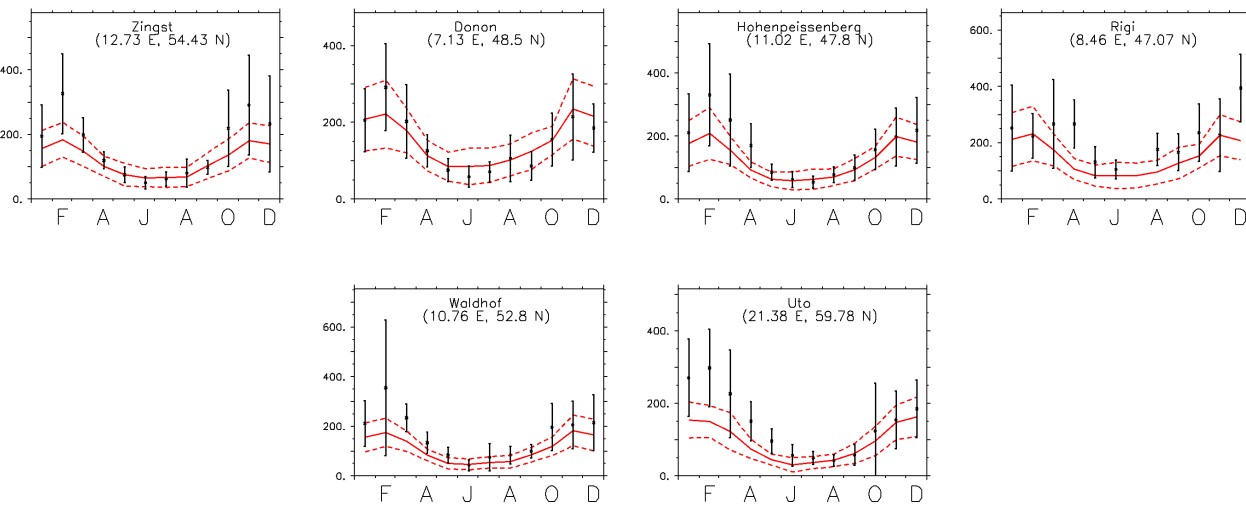

**Figure 2.** EMEP observations for benzene in the year 2005 (monthly average) in black, in red *RCP* scenario. Error bars show standard deviation of observations, red dashes are standard deviation of model simulation.

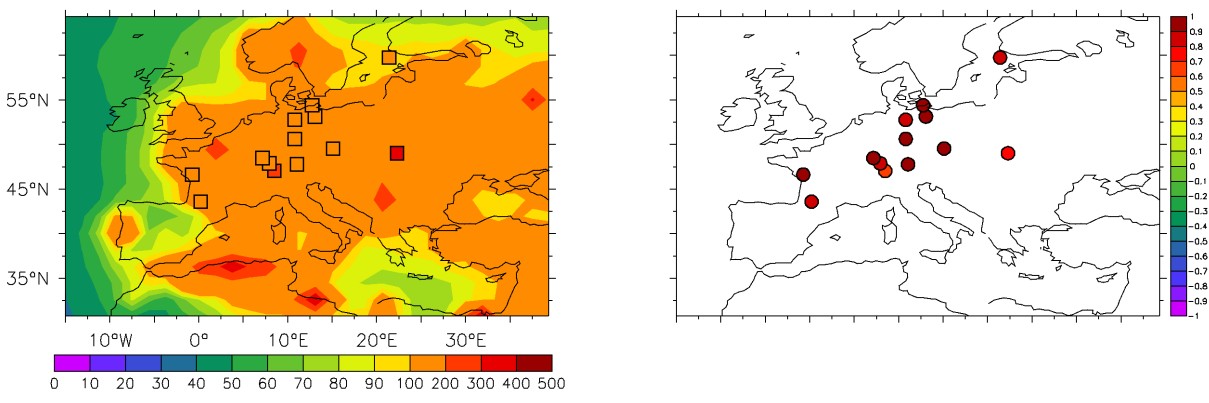

**Figure 3.** Top: annually averaged background mixing ratios of benzene for the *RCP* simulation, with the squares depicting annually averaged EMEP observations, both in $\mathrm{pmol/mol}$. Bottom: temporal correlation between observations and simulations for benzene.

previously observed in EMAC for other simpler VOCs (Pozzer et al., 2007). The *RCP* simulation yields an amplitude of the annual cycle that is closer to the observations than that of the *LIT* simulation (see Table 4). In addition, the observations show larger standard deviations than the simulations, although in the *RCP* case, this difference is relatively small (the ratio between the observed standard deviation and the *RCP* standard deviation is 1.48). The ratio RMS(*RCP*)/STD(OBS) is below 1, and the temporal correlation between the observations and the *RCP* scenario is very good (above 0.8 in most cases; Fig. 3). This



supports the good representation of the observations by the *RCP* simulation. On the other hand, the *RCP* simulation under-estimates the annual average systematically (35%) compared to the EMEP dataset, which is consistent with underestimation compared to the EEA dataset (34%).

*Literature*: Statistics cannot be calculated for these data, because in general the measurements were not performed in rural background areas and time spans of the studies are not suitable for comparison. Nevertheless, they are useful for a qualitative interpretation. In particular, this set of observations helps to better understand spatial performance of the model in regions of America, Africa and Asia (see Fig 4). Observations for 28 US cities (Baker et al., 2008) were taken during summer months in background locations (thus excluding New York, Philadelphia and Salt Lake city). Chicago and Detroit are strong industrialized cities, and therefore the model is biased low in both simulations. Despite the low resolution of the model, in the rest of US cities the simulation is clearly able to capture spatial gradients towards the urban areas. When comparing with the *RCP* simulation for the month of July, we find an underestimation of 48% on average. In China, benzene was measured in different areas (mainly residential, commercial, and industrial) in 48 cities during the winter season (Barletta et al., 2005). Both simulations reproduce the observed spatial gradient well, but strongly underestimate the mixing ratios, probably because the instruments were located close to sources. In general the *RCP* simulation is closer to the observations. At 6 different locations in the Sahara desert region, observations show mixing ratios on the order of tens of $pmol/mol$ for the winter months in background remote locations (Yassaa et al., 2011). The *RCP* simulation consistently reproduces those mixing ratios during the winter. Compared to the observations at a regional background station in South Africa (Jaars et al., 2014), the model shows again a low bias (88% lower than the observations) but within a reasonable range. The model is able to represent the mixing ratios peak in Rio de Janeiro (Brazil) (Martins et al., 2007).

*CARIBIC:* Model results have been sampled along the flight tracks, and observations within the 200-300 hPa levels are compared to the annual mean of the simulation in the 250 hPA level. The *LIT* simulation shows a stronger underestimation than *RCP*, with the *RCP* simulation underestimating tropospheric benzene mixing ratios by 20% and the *LIT* simulation by more than 100%. For the *RCP* scenario, the underestimations appears to be lower in the free troposphere than at the surface (*RCP* simulation underestimates observations of EMEP by 34% and EEA by 35%). Despite the large annual variability in benzene mixing ratios, the model is able to capture the gradients along the Africa and Europe-Brasil paths. For the North America-Europe-Asia tracks, the high variability of the measurements makes it hard to compare them with the simulations. In general, the model shows smaller spatial variability than the measurements and a RMS/STD ratio slightly above one, as for the *EEA* and *Literature* data.

## 4.2 Toluene

*EEA*: Toluene mixing ratios for the European stations used in this study show an average annual value of 240 $pmol/mol$. Compared to observations, model results from the *LIT* simulation show an overestimation of only 5%, while the model results from the *RCP* simulation have a high bias of 22%. Similar to the case of benzene, simulated mixing ratios have a smaller standard deviation than observations, meaning lower simulated spatial variability. The small RMS for the *LIT* simulation supports the good representation of the spatial gradients of the mixing ratios.





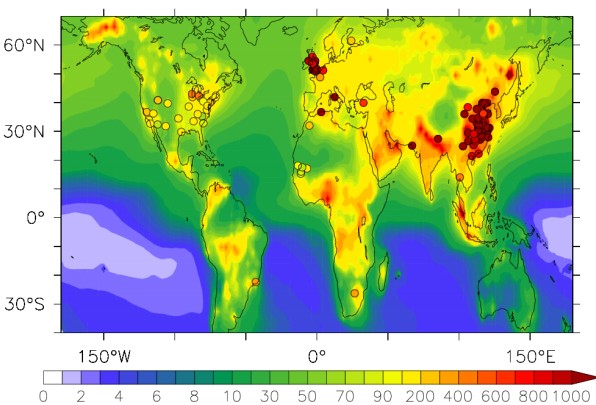

**Figure 4.** Annually averaged surface mixing ratios of benzene (pmol/mol) for the *RCP* simulation . Circles depict observations from literature.

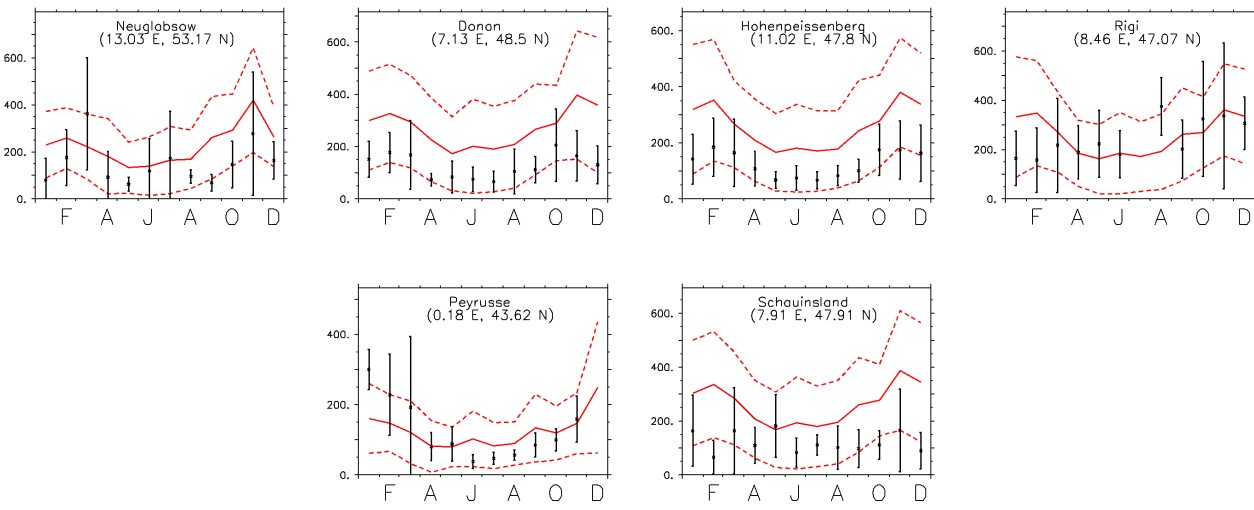

**Figure 5.** EMEP observations of toluene for 6 different locations for the year 2005 (monthly average) (in black) and the simulated toluene mixing ratios for the *LIT* simulation (in red). Error bars show standard deviation of the observations and red dashes show the standard deviation of the model simulation.

*EMEP*: Similar to the comparison with the EEA database, model results from the *LIT* simulation are closer to the EMEP observations than those from the *RCP* simulation, with annual average overestimations of 25% and 49%, respectively. Additionally, the RMS/STD ratio is for both simulations higher than in the comparison with the EEA database, which means that there is a lower spatio-temporal correlation. The temporal correlation for some stations is weaker than for benzene (see electronic supplement, doi:10.5194/acp-0-1-2016-supplement). Model results from both simulations reproduce the annual cycle;



in the case of the *LIT* simulation, the agreement is in general higher (see *LIT* case in Fig. 5). As opposed to benzene, the variability of the model results from the *LIT* simulation is 31% higher than that of the observations.

*Literature*: Generally, the performance of the model for the observations from the literature of toluene is similar to the one of benzene, and spatial gradients and large urban areas are correctly simulated. Nevertheless, for the same reasons as for benzene,
the model is showing a general underestimation for both simulations; *RCP* results are biased low by more than 3 times for benzene and almost 8 times for toluene. Both simulations capture the spatial distributions reasonably well, compared to the observations during summer for the US and during winter for the Sahara and China. The stronger discrepancies for toluene compared to benzene can be explained by the short lifetime of toluene in combination with the short distance from sources of the observations sites.

*CARIBIC*: In contrast to benzene, toluene annual average mixing ratios are underestimated greatly in both simulations, by more than 4 times (3.6 $\mathrm{pmol/mol}$ for observations and 0.8 $\mathrm{pmol/mol}$ for the *RCP* scenario). In contrast to the surface observations, the *RCP* simulation is closer to observations in the troposphere than the *LIT* simulation. Spatial gradients are best captured in the European-African and European-Asian tracks. Geographical variability is larger than for benzene, due to the shorter lifetime of toluene. In the simulations toluene is almost depleted above the planetary boundary layer, which suggests
a bad representation of the sinks. However, as pointed out by Helsel (1990), the underestimation due to the large number of measurements under the instrumental detection limit (1 $\mathrm{pmol/mol}$) is a source of error, since it artificially causes too high values in the observations. For the data used in this study, 46% of the CARIBIC observations for toluene are below detection limit, which partially explains the bias. As for benzene, the ratio RMS/STD is somewhat above one for both simulations.

## 4.3 Xylenes

*EEA*: Due to the low number of stations available for this dataset (only 2), the results may be not representative and therefore we did not include them in Table 4. However, for the two stations, mixing ratios from the *LIT* and *RCP* simulations are 66% and 100% higher than the observations, respectively.

*EMEP*: A comparison with model results of this set of 8 stations shows a similar result as the comparison for toluene. Figure 6 shows observations and model results from the *LIT* simulation. Results from both simulations are poorly correlated with
observations in terms of time and space (see electronic supplement, doi:10.5194/acp-0-1-2016-supplement). Model results from the *LIT* simulation are closer to the measurements than those from *RCP*, but in both cases the RMS/STD ratio is too high, which points at a low consistency in reproducing spatio-temporal features. EMEP observations are underestimated by 23% and 46% by results from the *LIT* and *RCP* simulations, respectively.

*Literature*: Observations of xylenes are only available for the US and for one location in China. As for other species, xylenes
are well represented in the US, with the exception of some cities. In China, the model reproduces the increase in mixing ratios towards the Hong Kong area. In the southern hemisphere, the model reproduces the polluted spots in South Africa and Rio de Janeiro.





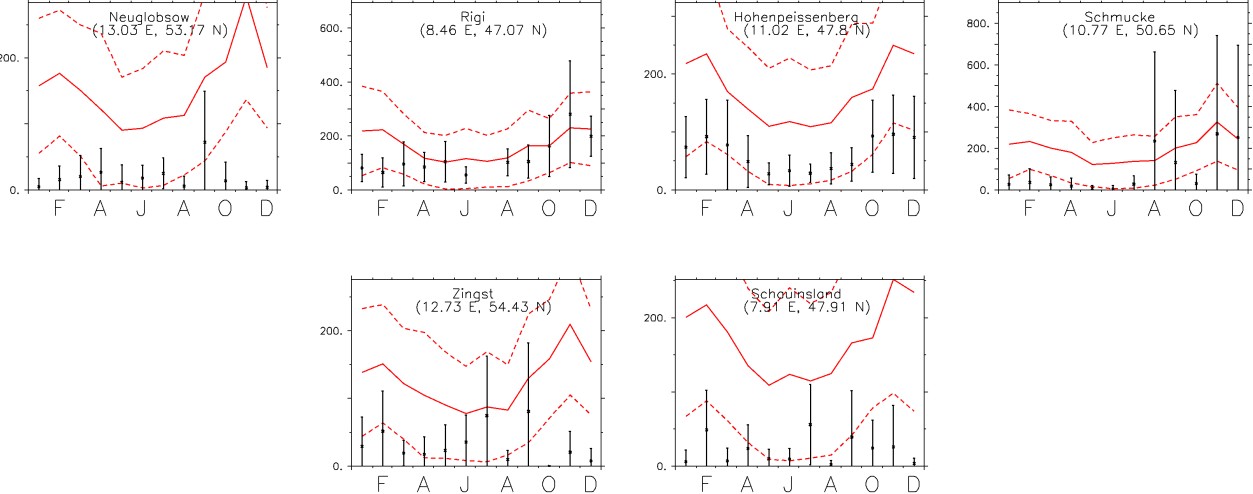

**Figure 6.** Mixing ratios of xylenes from EMEP observations in the year 2005 (monthly average) (in black), and from the *LIT* simulation (in red). Error bars show the standard deviation of the observations, red dashes show the standard deviation of the model simulation.

## 5 Global budget

Table 5 summarizes the global budget of aromatic compounds for the *RCP* simulation. This simulation has been selected because it reproduces benzene observations in terms of annual average mixing ratios, yearly cycle and spatial variations better than the *LIT* simulation. Moreover, the differences for toluene and xylenes are not significant between the two simulations.

For benzene, the total global primary emission of 8.5 Tg/yr is composed of anthropogenic emissions (81%), biomass burning emissions (19%) and chemical production (<1%). The sources are balanced by the sinks due to OH oxidation (87%), and wet and dry deposition (13%).

    Figure 7 shows modelled annually averaged mixing ratios of benzene, toluene and xylenes. For benzene (upper left panel), the surface mixing ratios are as high as 300-400 pmol/mol in highly urbanized and industrialized areas in the northern hemi-

10 sphere (US, Europa, China). Western Asia shows similar mixing ratios, probably due to the large petrol industry. The highest modelled mixing ratios can be found in India and China, due to large anthropogenic emissions. Central Africa and Northern Asia mixing ratios are mainly driven by biomass burning emissions. In general, areas with high mixing ratios of benzene are located close to sources, due to its relatively short lifetime. Over the oceans, mixing ratios vary between 20 and 70 pmol/mol (due to ship emissions). In southern hemispheric continental areas, we find mixing ratios in the 100-300 pmol/mol range. The

15 highest mixing ratios are found in Africa, due to the strong biomass burning season. Continental background areas show mixing ratios between 10-50 pmol/mol. Lowest mixing ratios (1-5 pmol/mol) are found in oceanic areas due to the far distances to sources.





**Table 5.** Global atmospheric budget of aromatic compounds from the RCP simulation. Units are Tg/yr in all cases, unless noted otherwise.

| | Benzene | Toluene | Xylene | Phenol | Styrene | Ethylbezene | TMB | Benzaldehyde |
|---|---|---|---|---|---|---|---|---|
| Sources (Tg/yr) | | | | | | | | |
| Biomass burning | 1.6 | 1.0 | 0.3 | 2.4 | 0.2 | 0.1 | 0.1 | 0.1 |
| Anthropogenic | 6.9 | 7.3 | 6.2 | 0.7 | 0.7 | 0.8 | 1.5 | 0.8 |
| Biogenic | | 0.4 | | | | | | |
| Chemical production | 0.01 | | | 4.4 | | | | 1.7 |
| Total Sources | 8.5 | 8.7 | 6.6 | 7.5 | 0.9 | 0.9 | 1.6 | 2.5 |
| Sinks (Tg/yr) | | | | | | | | |
| Dry Depostion | 1.5 | 1.2 | 0.7 | 0.2 | 0.0 | 0.1 | 0.2 | 0.1 |
| Oxidation by OH | 6.9 | 7.3 | 5.6 | 4.2 | 0.2 | 0.7 | 1.3 | 0.6 |
| Oxidation by NO3 | | | 0.1 | 2.3 | 0.3 | 0.0 | 0.0 | 0.1 |
| Oxidation by O3 | | | | | 0.1 | | | |
| Photolysis | | | | | | | | 1.9 |
| Total Sinks | 8.5 | 8.6 | 6.4 | 6.7 | 0.7 | 0.8 | 1.5 | 2.7 |
| Burden (Gg) | 231 | 62 | 22 | 3 | 0 | 5 | 3 | 2 |
| Lifetime (days) | 8.3 | 1.0 | 0.5 | 0.1 | 0.1 | 1.2 | 0.2 | 0.1 |

Figure 7 (upper right panel) shows the modelled annual zonal mean benzene mixing ratios. Note the strong north-south gradient and the averaged mixing ratios of 60-100 pmol/mol for the free troposphere. The highest values are found at the surface in the northern hemisphere.

Toluene emissions sum up to 8.8 Tg/yr, a number similar to that of benzene emissions. Anthropogenic emissions (84% of the total) play a larger role for this compound than the biomass burning emissions (11% of the total). Additionally, the model estimates 4% of emissions from biogenic sources. Sinks are dominated by OH oxidation (85%), and the remaining 15% is removed by dry deposition.

Mixing ratios at the surface are in the order of 20-200 pmol/mol in continental areas (Fig. 7, middle left panel), which are larger than for benzene for specific urban regions urban (Europe), due to large anthropogenic emissions. However, in the free troposphere we find low mixing ratios (a few pmol/mol), due to the short lifetime of toluene. Background areas (oceans, deserts) show surface mixing ratios below pmol/mol levels for the same reason.

More than 95% of xylenes are emitted by anthropogenic sources. Total emissions sump up 6.6 Tg/yr, from which 88% is removed by OH, 11% by dry deposition and the remainder (<1%) by reaction with $NO_3$. Very low mixing ratios are present at the surface in the southern hemisphere (Fig. 7, bottom left panel), except for a few specific locations (i.e. Indonesia, Nigeria, São Paulo in Brazil). In the free troposphere, mixing ratios are below pmol/mol levels in the southern hemisphere, even in the lowermost levels.



Phenol has a different distribution of sources compared to other aromatics. The main is source is the atmospheric oxidation of benzene with OH, which produces 4.4 Tg/yr (59% of total sources). The second important source of phenol is the primary emission from biomass burning, which represents 32% of the total emissions. Anthropogenic emissions are only 9% of the total. Nevertheless, mixing ratios of phenol in the atmosphere are low because of its short lifetime.

Emissions of benzaldehyde, styrene and trimethylbenzenes sum up to 5 Tg/yr. Their spatial patterns resemble those of toluene, with most emissions, and hence higher mixing ratios, located in the northern hemisphere. Nevertheless, their mixing ratios are almost one order of magnitude lower than those of toluene at the surface and in the free troposphere. For this reason, they have only been measured in few campaigns (Baker et al., 2008; Yurdakul et al., 2013).

Coherent with their main sink (i.e. reaction with OH) and the main region of their emission (i.e. the Northern hemisphere), the burden of most species shows a clear annual cycle, with higher mixing ratios in winter than in summer. As an exception, phenol and styrene have an annual cycle with a small amplitude. The specific pattern in their atmospheric burden is caused by their very short lifetimes and the relative high strength of the biomass burning emissions during fall.

The atmospheric aromatic burden sums up 0.3 Tg. Benzene contributes 70% to the total mass, toluene and xylenes 25% and the remaining species 5%.

Estimated lifetimes of aromatics are for most species in the order of a day or below (except for benzene, toluene and ethylbenzene), and can be found in Table 5. Estimated lifetimes are in line with values from the literature Atkinson (2000).

## 6 Conclusions

The 3-D atmospheric chemistry general circulation model EMAC and an ensemble of airborne and surface observations were used to evaluate our current understanding of the global atmospheric budget of monoaromatic compounds, including benzene, toluene, xylenes, phenol, styrene, ethylbenzene, trimethylbenzenes and benzaldehyde.

We extended the chemical mechanism of MECCA in order to accurately describe the chemical reactions of aromatic compounds. Emissions of simple aromatics were included in the model, considering biomass burning, anthropogenic activities and natural sources. As sinks, wet and dry deposition were included.

Simulations with two different sets of anthropogenic emissions were evaluated against observations. The comparison with surface and aircraft observations shows that for benzene, the model seems to underestimate mixing ratios consistently at the surface and in the free troposphere, while the spatial distributions and seasonal cycles are well reproduced. The model captures the spatial variability and averaged mixing ratios at the surface of toluene well, but it does not accurately reproduce the seasonal cycle and considerably underestimates mixing ratios in the free troposphere. This suggests an overestimation of the efficiency of the chemical removal processes, of which the chemical reaction with OH is the most important. The uncertainty of the rate constant for the reaction of toluene + OH is about $5.6 \pm 0.9 \times 10^{-12}$ cm$^3$molecule$^{-1}$s$^{-1}$ at 298K, which implies a 30% error on the chemical sink estimate. Additionally, the relative bias of the observations, due to the large number of observations below the instrumental detection limit of the aircraft measurements, can partially explain the disagreement in the upper troposphere. The model shows large temporal discrepancies for mixing ratios of xylenes, although they remain within an acceptable range.



Because of the low mixing ratios of some species (phenol, styrene, ethylbenzene, trimethylbenzenes and benzaldehydes) in the atmosphere, and the limitations of the present instrumentation, a model-measurement comparison was not possible for all species. Therefore, a wider array of samples would be helpful to asses the model's accuracy in remote regions and constrain the respective global atmospheric budgets.

The budget of aromatic compounds is characterized by a total emission rate of 37 $Tg/yr$. For most species, with the exception of phenol, anthropogenic emissions are the main source. Large emissions are located in industrialized and heavily populated areas, such as Asia and Europe. Emissions from biomass burning play a secondary role on the global scale, although they can be the strongest source of aromatics in specific areas such as Central Africa, South America and boreal areas.

    The chemical production generates 6 $Tg/yr$ of aromatics (mainly phenol), making them nearly ubiquitous. Biogenic emis-
10 sions form only a small fraction of the total toluene source (4%), although other studies suggest that this fraction could be larger (Sindelarova et al., 2014). Photochemical reaction with OH is the most important removal process of aromatics from the atmosphere, followed by dry deposition. As an exception, styrene and benzaldehyde also react with $O_3$ and $NO_3$, respectively, as their primary sink.

    Further studies focused radical production, ozone formation (Bloss et al., 2005a; Nehr et al., 2014) and general impact of
15 aromatics on atmospheric chemistry will be performed based on the mechanism developed in this study.

*Acknowledgements.* We want to thank Ruud Janssen for useful discussion. We also want to thank Angela Baker for providing the CARIBIC data. The authors wish also to acknowledge the use of the Ferret program for analysis and graphics in this paper. Ferret is a product of NOAA's Pacific Marine Environmental Laboratory (information is available at http://www.ferret.noaa.gov).



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





**Figure 7.** The left column shows annually averaged surface mixing ratios (pmol/mol) for the *RCP* simulation. The right column shows the annual zonal average. Upper plots show benzene, the middle toluene and the bottom plots show xylenes. Figures for other species can be found in the electronic supplement (doi:10.5194/acp-0-1-2016-supplement).