# Peer review of "Global atmospheric budget of simple monocyclic aromatic compounds"

_Atmospheric Chemistry and Physics, 2015_

## Referee Comment (RC1) · Anonymous Referee #2 · 28 Mar 2016

The paper "Global atmospheric budget of simple monocyclic aromatic compounds" presents a comprehensive study on the atmospheric budget of aromatics in the gas-phase. The authors described the implementation of aromatic compounds into the EMAC model in great details, in terms of their emissions, chemistry, and depositions. The paper also presented the validation of the simulation for aromatics using an ensemble of surface and aircraft observations. The paper is clearly written and easy to follow. The method is fully described. The conclusion is not surprising given our current understanding of the sources and sinks for these aromatic compounds. However, I do appreciate (and I think the community would too) the great effort the authors put into the implementation of these new species into models, and potentially the applications of such a model capability. Thus I think the paper could be published in the Atmospheric Chemistry and Physics (ACP), but it seems the manuscript as-is may suit

better for publication in the Geoscientific Model Development (GMD). The authors may consider the following suggestions to improve the quality of the paper:

1. The validation of the simulations has been focused on whether the model can reproduce the observed mixing ratios for aromatics. This is of course useful. A step forward for the validation is to test the model's capability for simulating the observed species: species ratios. For example, we have been using the benzene/toluene, or toluene/xylenes ratios as photochemical clocks to determine the age of air mass, because they are typically co-emitted from similar sources and they all have different photochemical lifetime. This could at least give some indications about how confidence the chemistry is in the model.

2. The speciation of xylenes and trimethylbenzenes, etc. The model lumps isomers and assumes equal proportions of single isomers. It is unclear how the rate coefficients are calculated for the lumped species. And, is it a good assumption to assume equal proportion of single isomers when we know some isomer dominates? Justification is needed here considering these species are typically more reactive than benzene and toluene thus are expected to have larger atmospheric impacts. Other two thoughts about the speciation: 1) how sensitive are observation techniques used in the validation to those isomers? 2) how do the emission inventories used here separate those isomers, and what are their assumptions when they lump species?

3. 'Anthropogenic emissions represent the largest source of aromatics' is not something really exciting, because this has been known for a long time. Can the authors provide more sectorial information about these anthropogenic emissions? For example, solvent usage has been considered as the largest source for toluene and (lumped) xylenes but not for benzene in the RETRO inventory. Can this work say something about the importance about the solvent usage? Another example, are emissions from vehicles still an important source for aromatics in urban and rural areas? Insights in such sectorial emissions could really improve the quality of the paper.

4. Can the authors say something quantitatively about the RCP emission inventory for benzene, toluene, and xylenes? Are they good? How good? Are there any regions that need to be improved based on the validation in the paper? What are the weaknesses of this emission inventory for aromatics?

5. Is it really necessary to simulate 666 reactions and 229 species in order to reach the conclusions of the paper? Do the authors have any recommendations for a simplified chemistry for model communities? What are the advantages of comprehensive descriptions about the chemical reactions? The authors need to expand the motivations about this.

6. Tables 1 and 2 in the supplement are not self-explanatory at all. They will need to be modified.

7. I suggest that the '2.3 Sinks ' should be renamed as '2.3 Scavenging and dry deposition', as '2.2 Chemistry' is considered a part of 'Sinks' too.

---

## Referee Comment (RC2) · Anonymous Referee #1 · 19 Apr 2016

This paper presents global model simulations and their evaluation with observations of aromatic compounds (primarily benzene, toluene, xylenes). Aromatics are a significant component of tropospheric chemistry and important ozone and aerosol precursors in urban areas. The chemistry scheme used to represent individual compounds and their oxidation products is a somewhat simplified version of the Master Chemical Mechanism, but is much more explicit than generally used in global models.

This paper is a valuable contribution to the literature and I recommend publication after addressing the comments and corrections given below.

Abstract and throughout paper: Emissions, burden and loss rates of total aromatics should be reported in TgC as the individual aromatic species have different molecular weights.

[Figure]

Abstract, l.4: the current wording implies the emissions are a result of the model simulation, where as the anthropogenic and biomass burning amounts are determined by the emissions inventory used. You might want to re-phrase that sentence.

p.3, l.5: Guenther et al., 2012 should be cited here, and other places throughout the paper, instead of Sindelarova. Whenever you are referring to a fundamental or general aspect of the MEGAN biogenic emissions. Sindelarova presents an application of the MEGAN model, but did not develop the model, or determine which species have biogenic emissions.

p.3, l.10: Replace "Besides, there is" with "In addition, there are"

p.3, l.24: Place 'e.g.' at the start of the reference list.

p.3, l.26-27: re-write - not clear currently if Henze looked at SOA or not.

p.3, l.29: give the chemical formulae of each compound.

Table 3: Units are g-species per kg-(dry matter burned)? state more explicitly. Would be helpful to also list totals as TgC/yr.

p.6, l.8: rewrite "does not present such amount" Do you mean doesn't include them, or doesn't indicate them separately?

p.6, l.15: Guenther et al., 2006 only presents isoprene emissions. Do you mean MEGANv2.1 (Guenther et al., 2012)?

Table 4: Observations are really the reference for the model, so it would be more appropriate to give the ratio of the models to the observations: MLIT/Mobs and MRCP/Mobs.

Figure 2: How are the model results compared to the mountain sites? Do you interpolate the model value to the pressure of the observation sites (correct way), or do you just use the surface model value (probably not correct - as the model probably does not resolve the topography of the mountain site).

Figure 4: I do not find this figure very informative - the model results in Asia are not visible. Small points could be used to indicate the obs. locations, then plot model vs obs in a scatter plot.

p.14, l.15: What do you mean by a "bad representation of the sinks"? That OH is too high? Explain further.

p.14, l.16-18: How do you treat the observations that are below the detection limit? Rewrite these sentences.

Fig. 6: The large difference at Hohenpeissenberg could be due elevation differences between model and obs. (see comment about Fig. 4).

p.15, l.7: HCs are not removed by wet deposition.

p.17, l.9: 'Coherent' -> 'Consistent'. I don't understand what the location of the emissions (NH) has to do with the seasonal cycle.

p.17, l.13: 'totals' instead of 'sums up [to]'

p.17, l.15: rewrite to: 'on the order of a day or less'

p.18, l3: 'asses' should be 'assess'

---

## Author Response (AR1)

Dear Dr. Butler,

here we have listed the changes applied to the manuscript, following the referees' suggestions. To facilitate the comparison with the ACPD published version, the text modifications are highlighted in the manuscript appended to this letter.

Correction made following comments to referee #1:

**Abstract and throughout paper: Emissions, burden and loss rates of total aromatics should be reported in TgC as the individual aromatic species have different molecular weights.**

The authors agree with the referee comment, as it will be helpful (more intuitive) for an easier comparison of fluxes. Therefore, we changed the units to TgC in the manuscript.

**Abstract, l.4: the current wording implies the emissions are a result of the model simulation, where as the anthropogenic and biomass burning amounts are determined by the emissions inventory used. You might want to re-phrase that sentence.**

We rephrased the sentence in order to clarify this aspect: "Anthropogenic emissions provided by RCP database represent the largest source of aromatics in the model ($\simeq$ 23 TgC/yr) and biomass burning from the GFAS inventory the second largest ($\simeq$ 5 TgC/yr). The simulated chemical production of aromatics accounts for $\simeq$ 5 TgC/yr."

**p.3, l.5: Guenther et al., 2012 should be cited here, and other places throughout the paper, instead of Sindelarova. Whenever you are referring to a fundamental or general aspect of the MEGAN biogenic emissions. Sindelarova presents an application of the MEGAN model, but did not develop the model, or determine which species have biogenic emissions.**

We added the citation Guenther et al., 2012. However, we think that citing Sindelarova et al, is also helpful for the reader, as it mentions the total emissions of toluene.

**p.3, l.10: Replace "Besides, there is" with "In addition, there are"**

We changed it.

**p.3, l.24: Place 'e.g.' at the start of the reference list.**

We changed it.

**p.3, l.26-27: re-write - not clear currently if Henze looked at SOA or not.**

The intention of Henze et al, is the quantification of the SOA formation by aromatics. We rephrased the sentence. "This work focuses on the gas phase chemistry of simple aromatics, hence neglecting any SOA production. Other global studies as (Henze et al, 2008) include SOA production as they were focused on the aerosol phase."

**p.3, l.29: give the chemical formulae of each compound.**

We added them in the revised manuscript.

**Table 3: Units are g-species per kg-(dry matter burned)? state more explicitly. Would be helpful to also list totals as TgC/yr.**

We clarified this in the revised manuscript. We express now the fluxes in TgC/yr.

**p.6, l.8: rewrite "does not present such amount" Do you mean doesn't include them, or doesn't indicate them separately?**

The sentence have been rewritten to clearly state that RCP includes open cooking emissions, although in lesser amount than in Johnson et at. (2013). "In the present study open cooking emissions are included within anthropogenic sources but the *RCP* database does not present such large phenol emissions."

**p.6, l.15: Guenther et al., 2006 only presents isoprene emissions. Do you mean MEGANv2.1 (Guenther et al., 2012)?**

We removed it and added the adequate one.

**Table 4: Observations are really the reference for the model, so it would be more appropriate to give the ratio of the models to the observations: MLIT/Mobs and MRCP/Mobs.**

We agree with the reviewer's interpretation and modified the table.

**Figure 2: How are the model results compared to the mountain sites? Do you interpolate the model value to the pressure of the observation sites (correct way), or do you just use the surface model value (probably not correct - as the model probably does not resolve the topography of the mountain site).**

We compared observations with the surface layer of the model, because the model altitude at the surface is similar to the altitude of most of the stations. Nevertheless, for a more accurate comparison, in the revised manuscript we made the comparison including the altitudes (we did for EMEP and EEA, not for LITERATURE as not always this information was available). Table 4, text and figures have been changed accordingly in the revised manuscript. Additionally we found an error in the calculations on the EMEP data. This have been corrected.

**Figure 4: I do not find this figure very informative - the model results in Asia are not visible. Small points could be used to indicate the obs. locations, then plot model vs obs in a scatter plot.**

We added an scatter plot next to the original figure in order to better compare observations and model mixing ratios. Nevertheless, as we mention in the manuscript, the purpose of this plot is merely qualitative due to the limitations in the data (observations).

**p.14, l.15: What do you mean by a "bad representation of the sinks"? That OH is too high? Explain further.**

As we mention in the conclusion "The uncertainty of the rate constant for the reaction of toluene + OH is about $5.6 \pm 0.9 \times 10^{-12}$ cm$^3$molecule$^{-1}$s$^{-1}$ at 298K, which implies a 30% error on the chemical sink estimate.". We substitute "bad" to "too strong model sinks in those regions" in order to correct this in the manuscript.

**p.14, l.16-18: How do you treat the observations that are below the detection limit? Rewrite these sentences.**

The sentence has been modified to make this clear. "In this case, 46% of the CARIBIC observations for toluene are below detection limit, which partially explains the bias. We only use the other 54% of the data for the calculations in table 4."

**Fig. 6: The large difference at Hohenpeissenberg could be due elevation differences between model and obs. (see comment about Fig. 4).**

Model height for the grid box where Hohenpeissenberg station is contained, has only few meters difference from the observed altitude. Thus, we exclude this reason as a source of the large underestimation found. Therefore, no change have been made in the manuscript.

**p.15, l.7: HCs are not removed by wet deposition.**

In the model wet deposition process is included for aromatic HCs. However the contribution by this process is negligible. We have now stated it clearly in the manuscript.

**p.17, l.9: 'Coherent' $\longrightarrow$ 'Consistent'. I don't understand what the location of the emissions (NH) has to do with the seasonal cycle.**

The global atmospheric burden is dominated by the northern hemisphere emissions, this explains the large concentrations observed in the NH in comparison with the SH. Therefore, increases in the atmospheric burden in winter on the NH due to the decrease on OH concentrations explain the changes in the global burden. The text in the manuscript has been changed accordingly.

**p.17, l.13: 'totals' instead of 'sums up [to]'**

We changed it.

**p.17, l.15: rewrite to: 'on the order of a day or less'**

We changed it.

**p.18, l3: 'asses' should be 'assess'**

We changed it.

Correction made following comments to referee #1:

**1. The validation of the simulations has been focused on whether the model can reproduce the observed mixing ratios for aromatics. This is of course useful. A step forward for the validation is to test the model's capability for simulating the observed species: species ratios. For example, we have been using the ben-**

zene/toluene, or toluene/xylenes ratios as photochemical clocks to determine the age of air mass, because they are typically co-emitted from similar sources and they all have different photochemical lifetime. This could at least give some indications about how confidence the chemistry is in the model.

We added in the manuscript (in the "evaluation with observations" section) a description of the model representations of the toluene/benzene and xylene/benzene ratios compared to the LITERATURE observations (as this set of observations cover the largest area). The related figures were added in the supplement material since they are not essential for the comprehension of the paper.

**2. The speciation of xylenes and trimethylbenzenes, etc. The model lumps isomers and assumes equal proportions of single isomers. It is unclear how the rate coefficients are calculated for the lumped species. And, is it a good assumption to assume equal proportion of single isomers when we know some isomer dominates? Justification is needed here considering these species are typically more reactive than benzene and toluene thus are expected to have larger atmospheric impacts. Other two thoughts about the speciation: 1) how sensitive are observation techniques used in the validation to those isomers? 2) how do the emission inventories used here separate those isomers, and what are their assumptions when they lump species?**

The assumption of the equal proportions for xylenes and trimethyl benzene isomers is justified by the lack of information on the emissions relative ratios. Additionally, the RCP VOC speciation does not contain any isomer speciation for xylenes. The rate constant of the three isomers (for both species) present differences below 70%. Thus, we averaged the rate constant (weighted by the branching ratios) and we suppose our error is within an acceptable range. A detailed description of the calculation of the reaction rates for each channel has been added to the supplement material.

Furthermore, there are numerous examples in the literature where full speciation/measurements of the isomers is not provided (e.g Lee et al., 2005; Derwent et al., 2014). Consequently for a consistent comparison between observation and model simulations, we consider our approach reasonable.

**3. 'Anthropogenic emissions represent the largest source of aromatics' is not something really exciting, because this has been known for a long time. Can the authors provide more sectorial information about these anthropogenic emissions? For example, solvent usage has been considered as the largest source for toluene and (lumped) xylenes but not for benzene in the RETRO inventory. Can this work say something about the importance about the solvent usage? Another example, are emissions from vehicles still an important source for aromatics in urban and rural areas? Insights in such sectorial emissions could really improve the quality of the paper.**

We added a paragraph in the anthropogenic emissions subsection describing the contribution of each sector to the total anthropogenic emissions. "[...]. When into the sectors provided by the RCP, we found for benzene 49% of the emissions are originated in the residential sector, followed by the energy sector (29%). In the case of toluene, emissions are evenly split for transportation, energy, solvents and residential. Xylenes emission are similarly distributed as for toluene, however solvents are the leading source with 30% of the emissions and residential only

7%. Trimethylbenzenes are abundantly emitted by the transportation sector (90%), as well as other aromatics (60%)."

**4. Can the authors say something quantitatively about the RCP emission inventory for benzene, toluene, and xylenes? Are they good? How good? Are there any regions that need to be improved based on the validation in the paper? What are the weaknesses of this emission inventory for aromatics?**

The number of observations used for this paper is limited for an exhaustive evaluation of the emission inventory and furthermore this is out of the scope of this paper. Nevertheless, we can state that the comparison of scenarios shows a better agreement with observation in the RCP case. Previous literature estimates seems to underestimate benzene emissions. In the case of toluene and xylene, the difference is less evident, as the emissions in both scenarios are not dramatically different.

**5. Is it really necessary to simulate 666 reactions and 229 species in order to reach the conclusions of the paper? Do the authors have any recommendations for a simplified chemistry for model communities? What are the advantages of comprehensive descriptions about the chemical reactions? The authors need to expand the motivations about this.**

On the one hand, the chemical mechanism should be as comprehensive as possible for an accurate description of atmospheric chemistry. On the other hand, it is of course necessary to simplify the mechanism for usage in long-term simulations with global models. To achieve this, we have started to work on skeletal mechanism generation based on the directed relation graph with error propagation (DRGEP) method. However, this is work in progress and not ready for inclusion into the current manuscript. This work serve as a basis for further studies atmospheric impacts of these compounds. The mechanism is planned to be used for future studies related to impacts on ozone, hydroxyl radical and other trace species. In the revised manuscript we add information about future studies with this chemical mechanism.

**6. Tables 1 and 2 in the supplement are not self-explanatory at all. They will need to be modified.**

Explanations for the Tables 1 and 2 have been added in the supplement material.

**7. I suggest that the '2.3 Sinks ' should be renamed as '2.3 Scavenging and dry deposition', as '2.2 Chemistry' is considered a part of 'Sinks' too.**

We changed the naming of the section. Nevertheless, "Chemistry" can either be a sink and/or a source or aromatic VOCs. For example, phenol main source is the oxidation from benzene, producing 4Tg/yr. Therefore, we believe that "Chemistry" should remain in an independent section from "Sinks" in order to avoid possible misleading.

Yours,
David Cabrera-Perez (on behalf of all co-authors)

**Global atmospheric budget of simple monocyclic aromatic compounds**

[revised manuscript text omitted]

In the *RCP* simulation,  23 TgC of aromatics are released into the atmosphere, which represents 18% of the total anthropogenic VOC emissions. In the *LIT* scenario,  16 TgC are emitted, and the aromatics represent 13% of the total anthropogenic VOC emissions. When looking into the sectors provided by the RCP, we found for benzene 49% of the emissions are originated in the residential sector, followed by the energy sector (29%). In the case of toluene, emissions are evenly split for transportation, energy, solvents and residential. Xylenes emission are similarly distributed as for toluene, however solvents are the leading source with 30% of the emissions and residential only 7%. Trimethylbenzenes are abundantly emitted by the transportation sector (90%), as well as other aromatics (60%).

**Biomass burning**

Biomass burning presents a large source of VOCs for the atmosphere (Lamarque et al., 2010). This contribution is represented by the BIOBURN submodel. BIOBURN calculates the emission fluxes, based on the Global Fire Assimilation System (GFAS) datasets (Kaiser et al., 2012). GFAS uses satellite retrievals of fire radiative power and transforms these into dry matter combustion rates. GFAS has a daily time resolution, and therefore seasonal variations can be observed. The dry matter combustion rates are used in the model in combination with biomass burning emission factors to estimate the biomass burning emissions of specific compounds.

**Table 3.** Biomass burning emission factors for the BIOBURN submodel. Emission factors are given in units of  g (species) / kg (dry matter burned). Last column present the global biomass burning emissions for the year 2005.

[revised manuscript text omitted]

to the shorter lifetime of toluene. In the simulations toluene is almost depleted above the planetary boundary layer, which suggests  too strong model sinks in those regions. However, as pointed out by Helsel (1990), the underestimation due to the large number of measurements under the instrumental detection limit (1 pmol/mol) is a source of error, since it artificially causes too high values in the observations. In this case, 46% of the

5  CARIBIC observations for toluene are below detection limit, which partially explains the bias. We only use the other 54% of the data for the calculations in table 4. As for benzene, the ratio RMS/STD is somewhat above one for both simulations.

**4.3  Xylenes**

*EEA*: Due to the low number of stations available for this dataset (only 2), the results may be not representative and therefore we did not include them in Table 4. However, for the two stations, mixing ratios from the *LIT* and *RCP* simulations are 66%

10  and 100% higher than the observations, respectively.

[revised manuscript text omitted]

The atmospheric aromatic burden  totals 0.3 TgC. Benzene contributes 70% to the total mass, toluene and xylenes 25% and the remaining species 5%.

Estimated lifetimes of aromatics are for most species  on the order of a day or  less (except for benzene, toluene and ethylbenzene), and can be found in Table 5. Estimated lifetimes are in line with values from the literature Atkinson (2000).

**6 Conclusions**

The 3-D atmospheric chemistry general circulation model EMAC and an ensemble of airborne and surface observations were used to evaluate our current understanding of the global atmospheric budget of monoaromatic compounds, including benzene, toluene, xylenes, phenol, styrene, ethylbenzene, trimethylbenzenes and benzaldehyde.

We extended the chemical mechanism of MECCA in order to accurately describe the chemical reactions of aromatic compounds. Emissions of simple aromatics were included in the model, considering biomass burning, anthropogenic activities and natural sources. As sinks, wet and dry deposition were included.

Simulations with two different sets of anthropogenic emissions were evaluated against observations. The comparison with surface and aircraft observations shows that for benzene, the model seems to underestimate mixing ratios consistently at the surface and in the free troposphere, while the spatial distributions and seasonal cycles are well reproduced. The model captures the spatial variability and averaged mixing ratios at the surface of toluene well, but it does not accurately reproduce the seasonal cycle and considerably underestimates mixing ratios in the free troposphere. This suggests an overestimation of the efficiency of the chemical removal processes, of which the chemical reaction with OH is the most important. The uncertainty of the rate constant for the reaction of $\mathrm{toluene} + \mathrm{OH}$ is about $5.6 \pm 0.9 \times 10^{-12}$ $\mathrm{cm^3 molecule^{-1} s^{-1}}$ at 298K, which implies a 30% error on the chemical sink estimate. Additionally, the relative bias of the observations, due to the large number of observations below the instrumental detection limit of the aircraft measurements, can partially explain the disagreement in the upper troposphere. The model shows large temporal discrepancies for mixing ratios of xylenes, although they remain within an acceptable range. We can conclude that the *RCP* scenario captures better total amounts of aromatics released to the atmosphere than the *LIT*. Nevertheless, in both scenarios we observed underestimation of the observations which could indicate an underestimation on the emission ratios.

[revised manuscript text omitted]